Climate change influences on the geographic distributional potential of the spotted fever vectors Amblyomma maculatum and Dermacentor andersoni

Alkishe Abdelghafar abduelkeesh8614@gmail.com 1 2
Peterson A. Townsend 1
1 Biodiversity Institute, University of Kansas , Lawrence , KS , United States of America
2 Zoology Department, Faculty of Science, Univeristy of Tripoli , Tripoli , Libya
Gillespie Joseph
Electronic publication date: 2022 May 3
Publication date: 2022
Volume: 10
Electronic Location ID: e13279
Received 2022 Jan 3; Accepted 2022 Mar 24
Copyright: ©2022 Alkishe and Peterson
Copyright year: 2022
Copyright holder: Alkishe and Peterson
License: This is an open access article distributed under the terms of the Creative Commons Attribution License, which permits unrestricted use, distribution, reproduction and adaptation in any medium and for any purpose provided that it is properly attributed. For attribution, the original author(s), title, publication source (PeerJ) and either DOI or URL of the article must be cited.
License URL: https://creativecommons.org/licenses/by/4.0/

Keywords: Gulf Coast tick, Rocky Mountain wood tick, Ecological niche modeling, Climate change, GCMs, RCPs, North America

Funding: The National Science Foundation OIA-1920946 This research was supported by a grant from the National Science Foundation (OIA-1920946). The funders had no role in study design, data collection and analysis, decision to publish, or preparation of the manuscript.

==============================
Amblyomma maculatum (Gulf Coast tick), and Dermacentor andersoni (Rocky Mountain wood tick) are two North American ticks that transmit spotted fevers associated Rickettsia. Amblyomma maculatum transmits Rickettsia parkeri and Francisella tularensis, while D. andersoni transmits R. rickettsii, Anaplasma marginale, Coltivirus (Colorado tick fever virus), and F. tularensis. Increases in temperature causes mild winters and more extreme dry periods during summers, which will affect tick populations in unknown ways. Here, we used ecological niche modeling (ENM) to assess the potential geographic distributions of these two medically important vector species in North America under current condition and then transfer those models to the future under different future climate scenarios with special interest in highlighting new potential expansion areas. Current model predictions for A. maculatum showed suitable areas across the southern and Midwest United States, and east coast, western and southern Mexico. For D. andersoni, our models showed broad suitable areas across northwestern United States. New potential for range expansions was anticipated for both tick species northward in response to climate change, extending across the Midwest and New England for A. maculatum, and still farther north into Canada for D. andersoni.

Introduction

Beside the tick Dermacentor variabilis, Amblyomma maculatum (Gulf Coast tick), and D. andersoni (Rocky Mountain wood tick) are three North American ticks that transmit spotted fever (Boorgula et al., 2020; CDC, 2018; CDC, 2019). Spotted fever rickettsioses (spotted fevers) are a group of bacterial pathogens that cause disease to humans by exposure to infected ticks or mites (CDC, 2019). In the United States, there are several spotted fevers: Rocky Mountain spotted fever (RMSF), which is the most documented spotted fever, caused by Rickettsia rickettsii; R. parkeri rickettsiosis caused by Rickettsia parkeri; rickettsialpox caused by R. akari; and Pacific Coast tick fever caused by R. philippi (CDC, 2019). Amblyomma maculatum transmits R. parkeri, and Francisella tularensis which cause diseases in humans, and Hepatozoon americanum, which causes health problems in dogs (Sonenshine, 2018). Dermacentor andersoni transmits R. rickettsii, Anaplasma marginale, Coltivirus (Colorado tick fever virus), and Francisella tularensis (Alkishe, Raghavan & Peterson, 2021; Dantas-Torres, Chomel & Otranto, 2012).

Amblyomma maculatum and D. andersoni have different geographic distributions: A. maculatum occurs throughout the southern states of the Gulf Coast and Mid-Atlantic states (Cumbie et al., 2020), whereas D. andersoni occurs throughout the Rocky Mountain region, Nevada, California, and southwestern Canada (British Columbia, Alberta, and Saskatchewan; Animal Diversity Web (ADW), 2021). Those different geographic ranges are associated with different climate conditions for each tick species (Fig. 1). Amblyomma maculatum is found in different months in different states with fall and winter considered as low-activity seasons for this species (Nadolny & Gaff, 2018), whereas D. andersoni is found in hot and dry areas in summer (Wilkinson, 1967).

Figure 1 Occurrence points and calibration areas for Amblyomma maculatum (blue dots and blue buffer) and Dermacentor andersoni (red dots and red buffer) in geographic and environmental space.

Climate warming is warming North America dramatically. Mean global temperature has increased more than 1°C owing to anthropogenic greenhouse gas emissions (Djalante, 2019). This increasing temperature has caused mild winters with increasing rain more than snow during winter, and more extreme drier periods during summers (Wuebbles et al., 2017). Increasing temperature can also affect vector disease survival, abundance, and activity as well as transmission dynamics, re-emergence of vector-borne diseases, and geographic expansions (Rocklöv & Dubrow, 2020).

Here, we used ecological niche modeling (ENM) to assess the geographic potential of these two medically important vectors of diseases in North America under current conditions and then transfer those models to the future under different scenarios, with special interest in highlighting potential range new expansion areas. We also assess the model uncertainty for projected future models to highlight areas with high versus low confidence of geographic expansions.

Methods

Data preparation

We obtained totals of 255 and 586 occurrence points for A. maculatum and D. andersoni, respectively. Those data were obtained from various sources: Global Biodiversity Information Facility (GBIF; http://www.gbif.org), VectorMap (http://vectormap.si.edu/), and BISON (https://bison.usgs.gov) (sources summarized in Supplemental Information 1). We followed Cobos et al. (2018) in cleaning the data to remove errors that clearly fall outside of the known geographic distribution of the species, duplicate records, and localities with missing or meaningless coordinates such as zero degrees latitude and zero degrees longitude (0°N, 0°E), or georeferencing errors (records in ocean and far from coast). We used the spTthin R package to reduce the data spatially based on a 50 km distance filter for several reasons: based on precision of the occurrence points in the area, environmental heterogeneity that present in the area, and to avoid problems with autocorrelation (Aiello-Lammens et al., 2015). In the end, we had 93 and 82 occurrence points for A. maculatum and D. andersoni, respectively. We divided the final occurrence data randomly into two sets: 50% for model calibration and 50% for evaluation steps involved in model calibration. For producing final models, we used the entire cleaned occurrence points.

Delineate the calibration area

The accessible area (M) is the set of places to which the species has had access over relevant time periods, and depends on the dispersal of the species from populations (Barve et al., 2011). Since the movement of tick species is associated with the movement of host species, we assumed ample dispersal abilities for the ticks (Nadolny & Gaff, 2018; Sonenshine, 2018). As such, we created 200 km buffer areas around the known occurrence points for each species (Fig. 1).

Environmental variables

For the current time, bioclimatic variables were downloaded from WorldClim version 1.4, at 10′ spatial resolution (Hijmans et al., 2005) (available at http://www.worldclim.org/). We removed variables 8, 9, 18, and 19 because of known spatial artefacts (Escobar, 2020). The 15 remaining variables were masked to the calibration area (M) for each species. We then used principal component analysis (PCA) to reduce dimensionality and multicollinearity among those variables. After having PCA results, we created 11 sets of environmental variables that represent all possible combinations of the first four principal components to test them with other parameter settings to choose best models during model calibration, following Cobos et al. (2019) (see below).

For future climatic conditions, we used five general circulation models (GCMs) under two representative concentration pathway scenarios (RCP 4.5, and RCP 8.5). Future climate data layers were downloaded from the Climate Change, Agriculture and Food Security (CCAFS) database at 10′ resolution (available at: http://www.ccafs-climate.org/data_spatial_downscaling). GCMs used were (1) National Center for Atmospheric Research (NCAR_CCSM4); (2) Met Office Hadley Centre (HadGEM2); (3) Model for Interdisciplinary Research on Climate (MIROC5); (4) Institut Pierre Simon Laplace (IPSL_CM5A); and (5) Russian Institute for Numerical Mathematics Climate Model Version 4 (INM_CM4). GCM choice was based on frequency of use in other such research applications, and on full availability of scenarios for both RCP scenarios.

Ecological niche modeling and model transfers

The combination of 11 sets of environmental variables, 15 feature classes (all combinations of linear = l, quadratic = q, product = p, hinge = h), and 17 regularization multiplier values (0.1 to 1 at intervals of 0.1, and 2 to 10 at intervals of 1) resulted in 2,805 candidate models for each species. We evaluated candidate models based on statistical significance (partial ROC, P ≤ 0.05; Peterson, Cobos & Jiménez-García, 2018), predictive performance (omission rates, <5%; Anderson, Lew & Peterson, 2003), and a criterion of minimum complexity (Akaike Information Criterion corrected for small sample sizes, AICc; Warren & Seifert, 2011). Specifically, we used differences between particular AICc values and the minimum values (ΔAICc < 2) to select best model parameter settings with which to produce final models.

Final models

For creating final models, we used the complete set of occurrences and the parameterizations selected during model calibration. We created 10 bootstrap replicates, and transferred the models across North America (Mexico, United States and Canada) under current and future scenarios. We calculated medians of all replicate medians from final predictions for each calibration area in which final models were produced to summarize model results. Then, we binarized models using a threshold of allowable omission error rate (E) of 5%, assuming that as a percentage of data may have included errors that misrepresented environments used by the species.

We calculated differences in suitable areas between current and the two future scenarios RCP (4.5, and 8.5). For representing changes of suitable areas, we used the agreement of changes (stable, gain, loss) among the five GCMs per RCP scenario. Simply, for each RCP scenario, we took all projections to future conditions based on distinct GCMs and compared against the current projection, and quantified the agreement of gain and loss of suitable areas, as well as the stability of suitable and unsuitable conditions. All modeling analysis steps were done in R 3.5.1 (R Core Team, 2018) using Maxent 3.4.1 (Phillips et al., 2017), implemented in the kuenm package (Cobos et al., 2019).

Uncertainty in model projections

We used the mobility-oriented parity metric (MOP, considering the nearest 5% of reference cloud) (Owens et al., 2013) to assess strict extrapolation risk. We also calculated variance arising from distinct sources (replicates, parameter settings, GCMs, and RCPs) in our model projections (Peterson, Papeş & Soberón , 2018). Both model variability and strict extrapolation were represented geographically following Owens et al. (2013) and Cobos et al. (2019), respectively.

Results

Model calibration results

From among 2,805 candidate models for each of A. maculatum and D. andersoni, 2,728 and 2,554 were significantly better than random expectations, respectively (pROC test, p ≤ 0.05). Of these models, 2,129 and 761 met the omission rate criteria, (i.e., OR ≤ 0.05) respectively. Based on AICc, 55 and two models were selected as best models for A. maculatum and D. andersoni, respectively. For A. maculatum, models performed better with the variables in set 1 (PC1, PC2, PC3, PC4), set 2 (PC1, PC2, PC3), set 3 (PC1, PC2, PC4), and set 6 (PC1, PC2), whereas for D. andersoni variables in sets 4 (PC1, PC3, PC4) and 7 (PC1, PC3).

Current and future potential distribution

Amblyomma maculatum

Current model predictions for A. maculatum showed suitable areas across the southern United States (Florida, Georgia, South and North Carolina, Virginia, West Virginia, Maryland, Delaware, Kentucky, Tennessee, Arkansas, Alabama, Mississippi, Louisiana, Oklahoma, and Texas), and in the Midwest (Missouri; eastern Kansas; southern Illinois, Indiana, and Ohio), and restricted areas of northeastern states (New Jersey and Pennsylvania). Suitable areas extend to include areas in western states (Arizona, California, Oregon, and Washington), although those areas are not likely accessible to the species (Fig. 2). Our models also showed suitable areas for the species across parts of eastern, western, and southern Mexico (Quintana Roo) (Fig. 2).

Figure 2 Left panel: potential suitable areas of Amblyomma maculatum based on binarized (5% threshold) models under current conditions (in blue and gray), and future (blue = no longer suitable, red = newly suitable) conditions.

Right panel: agreement in strict extrapolation areas among the five general circulation models. Results are presented for RCP 4.5 (top) and RCP 8.5 (bottom).

Future model transfers showed stable suitable areas (i.e., suitable in current time and in the future time) across the South, Midwest, and the Northeast, in the United States (Fig. 2). Areas of range reduction (loss) were in restricted areas in Kansas, Oklahoma, and Texas. Range expansion (gain) was anticipated in the northeastern (Pennsylvania, New York, Connecticut, Rhode Island, Massachusetts, Vermont, New Hampshire, Maine) and midwestern states (Kansas, Missouri, Nebraska, Iowa, Illinois, Indiana, Ohio, Michigan, Wisconsin) (Fig. 2). In general, we noted greater agreement among models in terms of losses and gains in the RCP 8.5 scenario compared to RCP 4.5.

Dermacentor andersoni

Current-time range predictions for D. andersoni showed broad suitable areas across Washington, Idaho, Oregon, California, Montana, Nevada, Utah, Wyoming, and Colorado, in cases where this species is known to occur in the United States. Climatically suitable areas extended across the Midwest, and Northeast and in some southeastern states (Fig. 3), although these areas are not likely accessible to the species. Currently suitable areas were also observed in parts of central and western Canada (British Columbia, Alberta, Saskatchewan, and restricted areas in Manitoba) (Fig. 3).

Figure 3 Left panel: potential suitable areas of Dermacentor andersoni based on binarized (5% threshold) models under current conditions (in blue and gray), and future (blue = no longer suitable, red = newly suitable) conditions.

Right panel: agreement in strict extrapolation areas among the five general circulation models. Results are presented for RCP 4.5 (top) and RCP 8.5 (bottom).

Future model transfers showed stable suitable areas across the states listed above, with some degree of reduction in suitable areas in the western states including much of Washington, Oregon, California, Nevada, Arizona, New Mexico, and Utah, and restricted areas in Colorado, Idaho, and Montana (Fig. 3). Predictions for the two RCP scenarios showed closely similar patterns of range stability, expansion, and loss, with more agreement among models in the RCP 8.5 scenario (Fig. 3).

Model uncertainty

MOP results for A. maculatum showed that strict extrapolative areas among future scenarios were concentrated in northern parts in North America, particularly in Canada, and in some restricted areas of the United States and southern Mexico (Fig. 2). Model variability results showed almost no variation coming from replicates and RCPs, but high contribution to variation from GCMs and parameter choice (Fig. S3).

In D. andersoni, we noted high agreement of strictly extrapolative areas in both southern and northern North America, and in lesser degree in the eastern United States and Canada (Fig. 3). High model variability came mainly from parameter choice in the eastern United States; we noted low variation deriving from GCMs, RCPs, and replicates (Fig. S4).

Discussion

The geographic distributions of A. maculatum and D. andersoni are much wider today than they were in the recent past. For example, A. maculatum has expanded its geographic range from the southeastern United States to become well established in the Northeast in Connecticut (Molaei et al., 2021), and in the Midwest in southern Illinois (Jolley, 2020). Beside the movement of tick adults for long distances via their hosts to new areas, immature A. maculatum can also access new areas with the help of migratory birds; larvae and nymphs can move thousands of miles during bird migratory seasons from the southern United States north to southern Canada (Florin et al., 2014; Teel et al., 2010). Cuervo et al. (2021) showed similar suitable ranges using current time predictors, and demonstrated that levels of niche conservatism differed among different members of A. maculatum group (A. tigrinum and A. triste).

This study is the first to assess the geographic distributions of the spotted fever vectors A. maculatum and D. andersoni in North America under current and future climate conditions. We included uncertainty analyses (MOP analysis and model variability) in our future model projections to detect areas with strict extrapolation, and to assess variation coming from multiple sources, such as different GCMs and RCPs. We considered only abiotic climatic variables such as temperature and precipitation as predictors that may influence the geographic distributions of those tick species.

Our models predicted that suitable areas for A. maculatum will remain stable in most southern and Midwestern states, whereas few reductions in suitable areas were anticipated only in western parts of Texas, Oklahoma, and Kansas (Fig. 2). Most importantly, our models predicted newly suitable areas northward in the United States successfully, to cover areas that were recently discovered to hold new populations in Connecticut and Illinois (Jolley, 2020; Molaei et al., 2021) (Fig. 4). For D. andersoni, our models showed broader suitable areas beyond its known range (from Washington state to Colorado). Midwestern and eastern states; however, most of the anticipated reduction in ranges were in areas not known to hold this tick species. Most of the anticipated expansions in range were in northward in Canada (Fig. 3).

Figure 4 Detail of Fig. 2, showing the most recent confirmed established populations of Amblyomma maculatum in counties of Illinois and Connecticut in the United States (light blue boundaries) (Jolley, 2020; Molaei et al., 2021).

Gray indicates suitable areas under current and future conditions. Red color indicates newly suitable areas with climate change.

Our projections suggested higher potential of A. maculatum to invade new areas outside its native range mainly in the southeastern United States. For D. andersoni, suitable areas were mostly in northern North America (United States and Canada). We also noted more extensive strict extrapolative areas for D. andersoni than A. maculatum, especially in the eastern United States, which suggested caution about interpreting those areas as suitable for D. andersoni (Fig. 3).

Several significant limitations and caveats regarding predictions emerging from ecological niche modeling that should be considered. First, a species faces dispersal limitations and biotic interactions that may prevent it from occupying the full suitable area that corresponds to its fundamental ecological niche. Second, the variation in spatial precision associated with different occurrence data records, which can cause problems for model results. Third, data availability in which biases in sampling in regions more than others can cause biases in model output (Peterson, 2014). All these points have been considered in the design of our methodology to achieve the most robust model possible.

In the United States, numbers of documented spotted fever cases have increased in recent years, especially in 2017, with 6248 new cases (CDC, 2021). Previous analyses have noted overlap between reported cases in some states and suitable areas for spotted fever vectors including Dermacentor variabilis (Alkishe, Raghavan & Peterson, 2021; Boorgula et al., 2020). Spotted fever case data collected by the Centers for Disease Control and Prevention, and used by Alkishe, Raghavan & Peterson (2021) however, were lacking in full detail on the type of pathogen and associated tick species, which made it difficult to interpret the source of the infection.

In summary, using ecological niche modeling allowed us to highlight suitable areas of two medically important tick species in North America. We showed the potential for expansion of those tick vectors into new areas that were not suitable in the past with emphasis on the newly discovered dispersal of A. maculatum to those newly suitable areas in Illinois and Connecticut. We also showed the uncertainty and variability that can come from projection models to different times and places. These results help to recognize the uncertainty and source of variability in predicting suitability.

Supplemental Information

Supplemental Information 1 Occurence points of Amblyomma maculatum and Deramcentor andersoni

Click here for additional data file.

Supplemental Information 2 Supplementary Figures and Tables

Click here for additional data file.

AA thanks Amel Dwebi and Yosr Alkishe for their support during this work. We also thank Gengping Zhu and one anonymous reviewer for their comments, which allowed us to improve the manuscript.

Additional Information and Declarations

Competing Interests

Author Contributions

Data Availability

The authors declare there are no competing interests.

Abdelghafar Alkishe conceived and designed the experiments, performed the experiments, analyzed the data, prepared figures and/or tables, authored or reviewed drafts of the paper, and approved the final draft.

A. Townsend Peterson conceived and designed the experiments, analyzed the data, authored or reviewed drafts of the paper, and approved the final draft.

The following information was supplied regarding data availability:

The raw data is available in the Supplemental File.

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
