# Peer review of "Climate change influences on the geographic distributional potential of the spotted fever vectors Amblyomma maculatum and Dermacentor andersoni"

_PeerJ, doi:10.7717/peerj.13279_

## Round 0.1 · original submission · Major Revisions

Dear Drs. Alkishe and Peterson:

Thanks for submitting your manuscript to PeerJ. I have now received two independent reviews of your work, and as you will see, the reviewers raised some concerns about the research. Despite this, this the reviewers are optimistic about your work and the potential impact it will have on research studying the impact of climate change on the geographic ranges of human disease vectors. Thus, I encourage you to revise your manuscript, accordingly, taking into account all of the concerns raised by both reviewers.

While the concerns of the reviewers are relatively minor, this is a major revision to ensure that the original reviewers have a chance to evaluate your responses to their concerns.

Importantly, please ensure your Materials and Methods are clearly stated. The methods should be clear, concise and repeatable. Please ensure this, and make sure all relevant information and references are provided.

There are many minor suggestions to improve the manuscript (typos, nuances, etc.).

I agree with many of the concerns of the reviewers, and thus feel that their suggestions should be adequately addressed before moving forward.

I look forward to seeing your revision, and thanks again for submitting your work to PeerJ.

Good luck with your revision,

-joe

·

Basic reporting

This is a good writing manuscript, that employed niche and distributional model techniques to estimate potential distributions of two vectors of spotted fevers under future climate. The manuscript incorporated recent advances in model development and validation, with climate change effects were estimated under different scenarios, and forecast associated uncertainty were explored …I think it merits acceptance and a fast publication.
I have some minor comments on the writing and style issue for further improvement.

Abstract:
Line 20: if Rickettsias is a genus name, it should be italicized. In the middle of this line, R. parkeri was first mentioned, but we don’t know whether the genus name is Rickettsias, as Rickettsias was not italicized, please check this out.
Line 28-32: I think it is not necessary to mention these states in the Abstract, they are too many and would distract reader, people not live in the U.S. might not familiar where are these states.

Introduction:
Line 47: pay attention where you first mention Rickettsia, and should be full name, after that it can be abbreviated.
Line 59-62: You are talking about distribution of these two vectors here, so it would good to cite Figure 1 here, see also my below comment on how to improve Figure 1.

Methods:
Line 86. I think it is better to put some information on (0,0), I know they are long and lat, but people might feel wield about these two “0” in brackets…or simply remove them?
Results:
Line 176-183: as I mentioned before, people not in the U.S. would not know where are these states, what if put abbreviations of these state name on Figure 1?

Discussion
I think it would be good to incorporate some discussion on common limitation of ENM in estimating vectors’ potential distribution, like climate niche conservatism, biotic interaction, and dispersal issue. For example, those blue areas identified as suitable habitat loss, but the tick might also survive there in future because of microclimate effect or adaption….my colleagues always complain that even we identified habitat unsuitable, but they still find the insect there, we should not underestimate insect evolve potential especially invasive species.

Comments on Figures:
Figure 1: I think it would be good to insert a scatter plot panel in this map, with the distribution (i.e. occurrence associate variables) of these two vectors in temperature and precipitations dimensions, with x axis is annual mean temperature and y is annual precipitation. It would demonstrate what you are talking the ecological preference of these two vectors in introduction in Line 59-62.
Figure 4: It would be good to insert a small overview panel in a corner of figure 4, to show reader where this map is showing…. It would be also good to mark “Illinois” and “Connecticut” states on this map, so people not in the US would know where these two states.

gengping.zhu@wsu.edu

Experimental design

Good, the manuscript incorporated recent advances in model development and validation, with many candidate models were calibrated.

Validity of the findings

Results are based on models that incorporated recent advances in niche and distributional modelling, they are robust and statistically sound.

Additional comments

None

Reviewer 2 ·

Basic reporting

In the case of English, I reserve comments. Two references would be missing that could notably improve the manuscript Pascoe et al. 2019 and Fernando Cuervo et al. 2021. In the case of the hypothesis, it is not well reflected in the text.

Experimental design

no comment. It seems to me that they meet the requirements.

Validity of the findings

no comment. These medically important species provide evidence to make better use of control measures and the information provided is very relevant.

Additional comments

Summary
In the Coltivirus species, add spp. I think that in the summary, they do not mention anything about the distribution of these species in Mexico.

Keywords, maybe we could add North America

Line 43. Must say are three of the ticks
Line 88. Only explain in better detail, the reason why they chose the distance of 50 km, (dispersion of the tick? Or some other behavioral reason?).
Line 91. In what way did you divide the data 50-50, did you use any procedure?
Line 99. Justify the use of the 200km buffer?
Line 104. The pixel size is very large. Is there a difference between using the 5 minute size and the 10 minute size?
Line 116. In the last update of March 2020, the new CMIP6 scenarios were made available, which difference exists between SSPs and RCPs; can we observe the same results?
Line 176. Although the geographic location in the United States is mentioned, the stable ecological niche in Mexico is not mentioned.
Line 232. I think that although Fernando Cuervo et al. 2021, does not show the results of the models, but it is very important to mention in relation to the dispersion of niche conservatism that occurs with A. triste and A. trigrinum; therefore, the part of migration would be important in the North American region.

---

## Round 0.2 · Minor Revisions

Dear Drs. Alkishe and Peterson:

Thanks for revising your manuscript. The reviewers are very satisfied with your revision (as am I). Great! However, there are a few concerns to address. Please attend to these issues ASAP so we may move towards acceptance of your work.

Best,

-joe

·

Basic reporting

The authors have addressed all my concerns, current revisions look good to me, indeed the manuscript is much improved in this revision, either in writing or figures, which clear demonstrated ecological dimensions and potential distribution, at present under future climate scenarios.
I think it can be considered for publication.

Experimental design

None

Validity of the findings

None

Additional comments

None

Reviewer 2 ·

Basic reporting

The manuscript has many improvements throughout the document. A few small comments are added to the review. The manuscript is ready to be accepted in its form with the suggested modifications.

Line 97. I understand the tick dispersal part and the host part, but I would like you to add the reference to support that argument.

Line 182: From the southeast of Mexico, only mention the state of Quintana Roo

Line 203: match the word west ern

Fig1. The word Dermacentor is misspelled.

Experimental design

The manuscript has many improvements throughout the document. A few small comments are added to the review. The manuscript is ready to be accepted in its form with the suggested modifications.

Line 97. I understand the tick dispersal part and the host part, but I would like you to add the reference to support that argument.

Line 182: From the southeast of Mexico, only mention the state of Quintana Roo

Line 203: match the word west ern

Fig1. The word Dermacentor is misspelled.

Validity of the findings

The manuscript has many improvements throughout the document. A few small comments are added to the review. The manuscript is ready to be accepted in its form with the suggested modifications.

Line 97. I understand the tick dispersal part and the host part, but I would like you to add the reference to support that argument.

Line 182: From the southeast of Mexico, only mention the state of Quintana Roo

Line 203: match the word west ern

Fig1. The word Dermacentor is misspelled.

---

## Round 0.3 · accepted · Accept

Dear Drs. Alkishe and Peterson:

Thanks for revising your manuscript based on the concerns raised by the reviewers. I now believe that your manuscript is suitable for publication. Congratulations! I look forward to seeing this work in print, and I anticipate it being an important resource for groups studying the impact of climate change on the geographic ranges of human disease vectors. Thanks again for choosing PeerJ to publish such important work.

Best,

-joe